# A Network Traffic Anomaly Detection Method Based on Gaussian Mixture Model

Bin Yu [1], Yongzheng Zhang [1], Wenshu Xie [2], Wenjia Zuo [2], Yiming Zhao [2] and Yuliang Wei [1,*]

[1] School of Computer Science and Technology, Harbin Institute of Technology, Weihai 264209, China; 22s130455@stu.hit.edu.cn (B.Y.)

[2] China Academy of Launch Vehicle Technology, Beijing 100076, China

* Correspondence: wei.yl@hit.edu.cn

**Abstract:** How can we learn the normal behavior of some communication processes and predict whether a single communication is under attack, with massive network traffic data representing the time costs of each stage in a single communication process? This paper introduces a statistical method for detecting network traffic anomalies using the Gaussian mixture model. There are two aspects to our contributions. First, we show how to learn the normal behavior of a communication process under the assumption that its time costs are generated from the Gaussian mixture model. Secondly, we show that with the learned Gaussian mixture model, we can predict whether a data point is under attack by computing the likelihood that the data point is drawn from the learned Gaussian distribution. The experimental results show that our method reached high accuracy in some cases, while in some other cases that are more complicated, the data point may have more factors and cannot be represented simply by only one Gaussian mixture model.

**Keywords:** traffic data; anomaly detection; Gaussian mixture model

## 1. Introduction

Anomaly detection aims to find patterns in the data that do not perform the expected behavior [1]. It has been a topic of study for researchers since the early nineteen century. Their efforts have led to the development of a wide range of techniques, from statistical models to evolutionary computing approaches. Unfortunately, identifying and classifying all existing anomaly-detection techniques can be challenging due to the complexity of the topic. Many factors must be considered, such as the types of anomalies and systems being studied, the techniques and algorithms used, and technical challenges such as processing costs and network complexity. This can lead to fragmented literature on anomaly detection, with many works attempting to summarize the field but failing to provide a comprehensive overview of the full spectrum of anomaly-detection techniques. They are helpful in various applications, such as fraud detection for credit cards, intrusion detection for cyber-security, and fault detection in safety-critical cards. When dealing with massive data, how can we conduct anomaly detection? These techniques are classification-based, clustering-based, nearest neighbor-based, statistical, information-theoretic, and spectral. Eskin [2] proposed a mixture model to detect anomalies. The authors conjectured that the set of calls to the system with a probability of $1 - \lambda$ is normal and that intrusions have the probability of $\lambda$. The two data-generating distributions are called the majority (M) and the anomalous (A) distributions [3].

Many network communications initiated by pairs of nodes exist in the real world at any time. The network communication between two nodes is shown in Figure 1 and contains a client and a server, both of which interact with data according to some defined protocol. However, many network communications can be subject to a variety of network attacks, posing a security threat to both sides of the communication. Figure 2 illustrates a communication scenario that is subject to a man-in-the-middle attack [4]. While the advent

of 5G networks has driven dramatic changes in current cyberspace technologies, it has also placed higher security requirements [5]. Discovering anomalous network communications can help us identify potential network security hazards. Therefore, detecting anomalous network communications remains a vital task today.

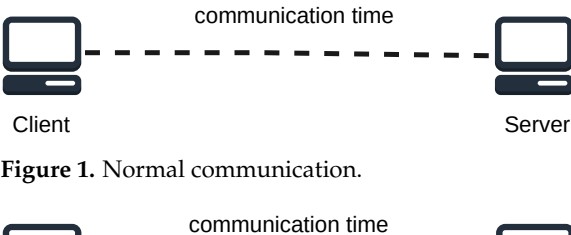

**Figure 1.** Normal communication.

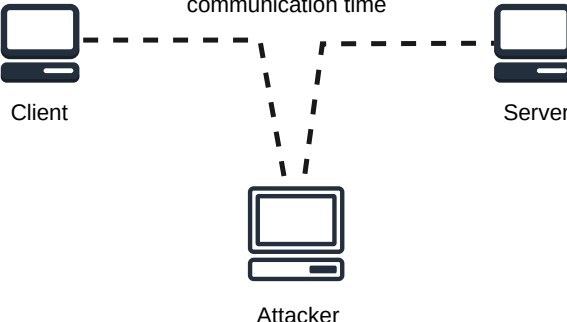

**Figure 2.** Abnormal communication.

We introduce a statistical traffic analysis method based on the Gaussian mixture model in this paper, which can detect whether anomalies such as man-in-the-middle attacks occur in a communication process based on the traffic communication time. In the modern network environment, a communication process is relatively stable under normal circumstances, meaning that its communication time tends to converge to a more fixed interval of values. However, when there is a deviation in the elapsed time of the traffic generated by a specific communication from the trend presented by the historical records, it is reasonable to suspect that there may be anomalies in this process, such as possible man-in-the-middle attacks. First, we assumed all traffic data are generated from a Gaussian mixture model, whose parameters we can learn through the EM algorithm [6]. Second, we show that with the learned parameters, we can predict whether an anomaly occurred in the communication process represented by a data point by calculating the probability of that data point coming from the learned Gaussian distribution.

## 2. Related Work

In terms of statistical methods, Callegari et al. [7] proposed a wavelet-based technique able to detect network anomalies with little time delay. They combined the use of sketches and wavelet analysis to reveal the anomalies in data collected at the router level. Additionally, they proposed a multiple time-scale analysis to improve the detection rate. In [8], Fernandes et al. introduced PCADS-AD, an autonomous profile-based anomaly detection system that uses dimensionality reduction and a principal component analysis system based on a dimensionality reduction process and principal component analysis. Miao Xie [9] developed a method for performing anomaly detection on a segment-by-segment basis. This method involves using random variables to analyze the predictability of neighboring data segments and then determining which segments exhibit abnormal behavior. In [10], Bang et al. introduced a hidden semi-Markov model (HsMM)-based intrusion detection system (IDS) that is designed to detect advanced LTE signaling attacks on wireless sensor networks (WSNs). This IDS uses the HsMM to analyze network traffic and identify anomalous behavior that may indicate an attack.

In terms of clustering methods, Carvalho et al. [11] proposed a proactive network-monitoring system that can automatically detect unusual events and reduce the need for

manual intervention and the potential for errors in decision-making. Their system involves creating a network profile called DSNSF (digital signature of network segment using flow analysis), which describes normal network usage using a clustering approach based on the ant colony optimization (ACO) metaheuristic. The modified ACO algorithm, called ACODS, is able to effectively analyze high-dimensional network traffic data and extract behavioral patterns through an unsupervised learning mechanism. To detect anomalous behavior, the system uses a pattern-matching technique called dynamic time warping (DTW). Bigdeli et al. [12] proposed a two-layer cluster-based structure for anomaly detection that aims to address some of the key limitations of existing methods. These limitations include the absence of labelled data, the difficulty in identifying new unknown anomaly patterns, the presence of noisy data, and the problem of high false alarm rates. The proposed system uses a Gaussian mixture model to cluster network data and represent these clusters in a way that allows the model to categorize new instances and ignore redundant ones. To address the issue of high false alarm rates, the system also uses a collective labelling method that labels new incoming instances in both a collective and incremental manner.

In terms of classification-based methods, Swarnkar and Hubballi [13] developed OCPAD, a method for detecting suspicious payload content in network packets using a one-class Naive Bayes classifier and frequency information of short sequences. This method allows for accurate anomaly detection in payloads. Kabir et al. [14] proposed an IDS based on the LS-SVM, a modified version of the standard SVM classifier that is more sensitive to outliers and noise in the training dataset. This makes it more effective at detecting anomalies in network traffic. Ashfaq et al. [15] proposed a semi-supervised learning approach that uses the NNRW classifier and calculates the fuzziness of unlabeled samples to improve its performance. The model discovers relationships between the output fuzzy membership vectors and misclassification rates to improve its ability to classify data accurately. Sornsuwit and Jaiyen [16] developed a novel ensemble approach for intrusion detection that uses the AdaBoost algorithm to combine the outputs of several different classifiers, including naive Bayes, decision tree, multi-layer perceptron (MLP), k-NN, and SVM. This approach allows the system to take advantage of the strengths of each classifier and improve the overall accuracy of the detection.

There are some other methods. David et al. [17] proposed a method for detecting DDoS attacks using fast entropy and flow-based analysis. Instead of counting the packets of each connection, their method aggregates the flows into a single one and considers the flow count of each connection at a given time interval. Bamakan et al. [18] proposed a new intrusion detection framework that uses a modified version of particle swarm optimization called time-varying chaos particle swarm optimization (TVCPSO). This adaptive and precise optimization method is used for parameter setting and feature selection for multiple criteria linear programming (MCLP) and support vector machines (SVM) at the same time.

## 3. Gaussian Mixture Model

A Gaussian mixture model (GMM) is a probabilistic model that assumes that all data points are generated from a mixture of a finite number of Gaussian distributions, with parameters that are not known to be known [19]. If we call each Gaussian distribution present in a GMM component, then the final GMM consisting of K components can be written as follows:

$$P(y|\mu_1, ..., \mu_K, \sigma_1^2, .., \sigma_K^2, \alpha_1, ..., \alpha_K) = \sum_{j=1}^{k} \alpha_j \mathcal{N}(y|\mu_j, \sigma_j^2), \tag{1}$$

where $y$ are the observed data points, $\mu_k$ are the means, $\sigma_j^2$ are the variances, $\alpha_j$ are the mixing proportions (which must be positive and sum to one), and $\mathcal{N}$ is a normalized Gaussian with specified mean and variance. It can be seen that a Gaussian mixture model can be considered as obtained by accumulating several different Gaussian distributions according to their corresponding weights.

A GMM is a generative model that considers a data point to be generated from one of the Gaussian distributions. According to the idea of GMM, the process of generating a data point can be expressed as follows: first, a distribution is randomly selected from all Gaussian distributions according to their weights, and then a data point is randomly sampled according to the mean and variance of this Gaussian distribution.

If all the parameters of a GMM are available, we can calculate the probability that a data point comes from each Gaussian distribution of this GMM. In the approach proposed in this paper, we estimate the parameters of the GMM and then find anomalous network traffic by calculating these probabilities and comparing them.

## 4. Method

### 4.1. Overview

Our method aims to detect the anomaly attack in the traffic data. Our training data in this paper represents the time cost of a single communication consisting of multiple stages. Each stage's time cost is also composed of different parts. However, each is affected by several factors, including the encryption algorithms for that stage, the link used for that communication, and the response time of the server. However, those few communications that suffer from the attack spend time with a statistical trend that significantly deviates from the historical data. Based on this phenomenon, each part's time cost is assumed to be generated from a Gaussian distribution. The parameters are determined by the corresponding factors that may affect the communication time cost. A Gaussian distribution reflects that most communication times tend to converge to a numerical interval. In the subsequent Section 5.1, we will demonstrate this phenomenon.

Since the sum of two Gaussian distributions remains a Gaussian distribution, it does not matter how many factors a stage is affected by, but the total time cost of a stage remains a Gaussian distribution. We set the number of links as the latent variable representing the Gaussian distribution the data point comes from in a mixture model. It is natural because there are multiple links for a communication stage to choose from, while there is only one choice concerning other factors. For example, there is only one certain encryption algorithm for one communication stage. To summarize, as shown in Figure 3, the time cost in each stage comes from a Gaussian mixture distribution. As a result, the number of Gaussian mixture models equals the number of different stages. The latent variable in a Gaussian mixture model represents the link used for that stage, i.e., the number of mixture components corresponds to the number of links.

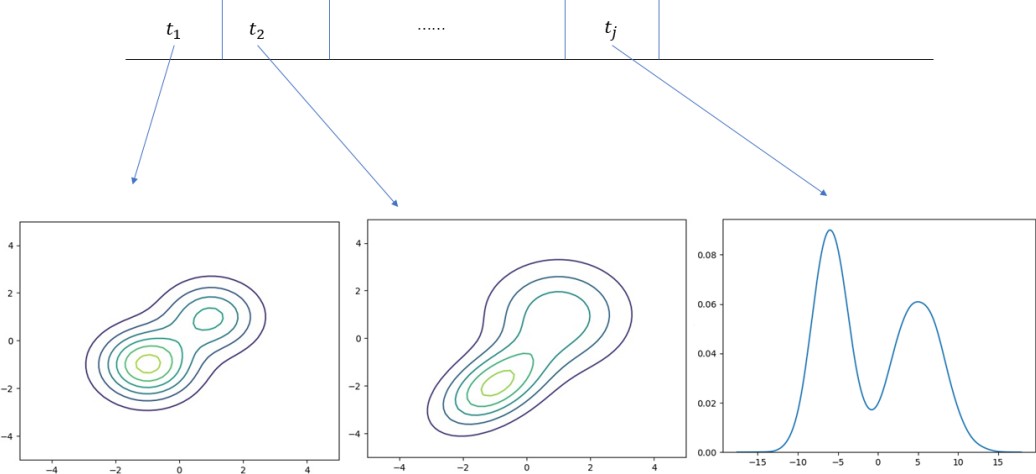

**Figure 3.** An example of a data point $x = (t_1, t_2, \ldots)$, which represents the complete communication process between the communicating parties and consists of several communication stages. $t_j$ represents the time cost in communication stage $j$. Each $t_j$ comes from a specific Gaussian mixture distribution, and the bottom half of the figure above plots this possible mixture distribution.

Our task is to compute the number of links in each stage and to work out parameters in each Gaussian distribution through our training data. In that way, we can learn the normal behavior of traffic data. Once a new piece of data arrives, we will be able to identify whether there an anomaly attack exists by computing the probability of the data piece coming from our learned Gaussian mixture model. We believe an anomaly attack occurred if the result is below the threshold set manually.

### 4.2. Parameter Estimation

For each data point in a Gaussian mixture model, the specific distribution it is generated from is unknown. We take the distribution from whence a data point comes as the latent variable. Due to the latent variables and the fact that each distribution has unknown parameters, the likelihood of data points is intractable to compute in the Gaussian mixture model. We use the expectation-maximization (EM) algorithm [20] to estimate the parameters. The EM algorithm is an iterative method for finding the maximum likelihood or maximum posterior estimates of the parameters, where the model is dependent on latent unobserved variables [21].

The core of the EM algorithm consists of two steps: expectation-step (E-step) and maximization-step (M-step). E-step estimates the value of parameters based on the current data and model and then uses this estimation value to calculate the expectation of likelihood denoted as a function $\mathcal{Q}$. M-step calculates the maximum likelihood estimation of function $\mathcal{Q}$ to determine the parameters. Two phases are executed alternately. EM algorithm ensures that the likelihood function $\mathcal{Q}$ increases after each iteration, so the likelihood will eventually converge, and we will learn the value of the parameters. The general process of the EM algorithm is as follows:

1. Determines what the latent variable is
2. E-step: to determine the form of the function $\mathcal{Q}$
3. M-step: to maximize the function $\mathcal{Q}$ to get the new parameter estimation for E-step

In the iterative process of the EM algorithm above, there is one function that is crucial: the function $Q$. The function $Q$ depends on the parameters of the model and the current round estimate of these parameters, so the function $Q$ is different for each round. In solving the problem of estimating the model parameters, the parameter variables of the model are given, and during the iterative process, we know the estimated values of these parameters for the current round. We can therefore write the specific form of the function $Q$ for the current round based on the definition. The expression of the function $Q$ is specified as follows:

$$Q(\theta, \theta^{(i)}) = E_Z[\log P(Y, Z|\theta)|Y, \theta^{(i)}], \tag{2}$$

where $\theta$ are the parameters of the model, $\theta^{(i)}$ are the current round estimate of these parameters, $Z$ are the latent variables, and $Y$ are the observed data points.

Through multiple iterations, the parameter values estimated by the EM algorithm at M-step converge gradually, and finally our parameter estimation results can be obtained.

The selection of the model and the estimation of the model parameters have become clear. The remaining steps can be done using the machine learning library Sklearn.

### 4.3. GMM-Based Anomaly Detection

Having known the value of the parameters in a Gaussian mixture model, given a new data point, we can calculate the probability that the data point comes from a particular Gaussian distribution. We believe an anomaly attack occurred if the result was less than the threshold. Take data point $x = (t_1, t_2, t_3)$, for example, suppose we already learned where the parameter of the distribution $x$ comes from, which is

$$t_1 \sim \mathcal{N}(\mu_1, \sigma_1^2), t_2 \sim \mathcal{N}(\mu_2, \sigma_2^2), t_3 \sim \mathcal{N}(\mu_3, \sigma_3^2). \tag{3}$$

There is a new data point $y = (\widetilde{t}_1, \widetilde{t}_2, \widetilde{t}_3)$. Data in each stage are independent. So, we can calculate

$$p(y) = p(\widetilde{t}_1) \times p(\widetilde{t}_2) \times p(\widetilde{t}_3), \tag{4}$$

where $p(y)$ denotes the probability $y$ comes from the same distribution as $x$, $p(\widetilde{t}_i)$, $i = 1, 2, 3$ denotes the probability, and $\widetilde{t}_i$ comes from $\mathcal{N}(\mu_i, \sigma_i^2)$ like $t_i$ does .

We think $y$ does not come from the same distribution as $x$ if $p(y) < threshold$, i.e., an anomaly attack is detected.

## 5. Experiments and Results

To begin with, we assumed that all communication steps share the same links. As it turns out, this assumption is not quite correct. Secondly, we assumed that different steps use different links and separately learn the parameters. Then, we observed how the number of links, i.e., the number of mixture components, affects the model's behavior. Figure 4 is the behavior of each model as the number of links varied.

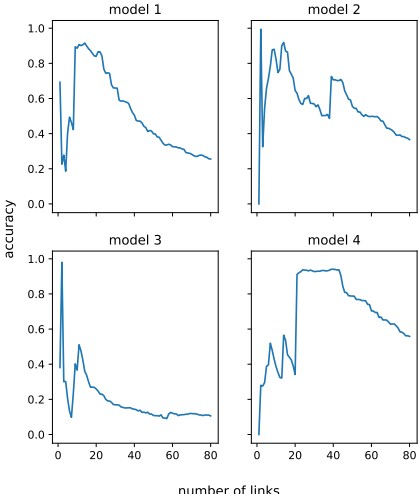

**Figure 4.** Each model's accuracy changes as the number of links varies.

### 5.1. Dataset

First, we introduce the scenario in which the dataset is generated. As shown in Figure 1, a typical network session consists of a client and a server. Depending on the session protocol, the client and the server may have multiple network communication processes. In our work, both sides of one communication use the IKEv2 (Internet Key Exchange version 2) protocol. The IKEv2 protocol is used to negotiate keys, and the complete protocol interaction process generates six network communication stages [22]. The round-trip time (RTT) of these network traffic communications can be measured by opening a packet-catching program for network packets on the client side. The actual RTT may vary for each communication, and the exact time may be affected by, for example, the algorithm's time complexity, the CPU load, and the degree of network congestion. We can obtain a dataset containing many samples by recording the RTT value for each network communication. We expect that we can use this dataset to fit multiple Gaussian mixture models and find anomalous traffic communication processes from them. For example, Figure 2 shows an anomalous traffic communication process in which both sides suffer from a man-in-the-middle attack.

We use a real-world dataset with 4182 rows. Each row represents one communication session, consisting of 6 columns, i.e., 6 stages. Each column represents the RTT time of a communication stage. There are three stages using the same encryption algorithm, so they can be thought of as being drawn from the same Gaussian mixture distribution. Thus, we have only four Gaussian mixture models to calculate. Figure 5 is a visualization of the data points of different stages. The RTT times of different stages show different

numerical characteristics because, according to the communication protocol, different tasks need to be performed by the communicating parties at different phases. In one fixed stage, the communication time of the sample is concentrated in a specific period under the influence of the noise in the communication process.

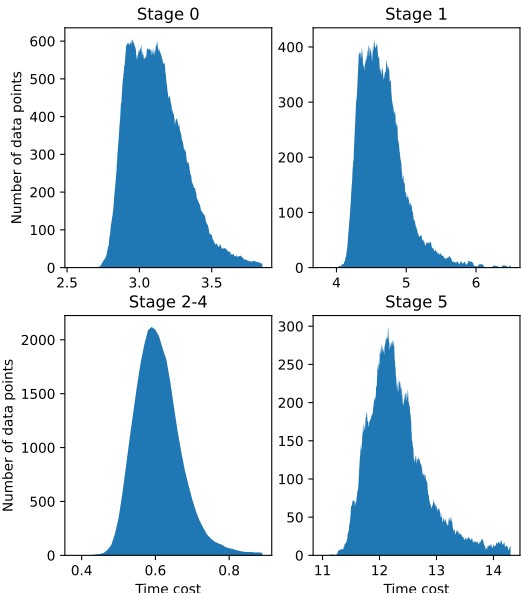

**Figure 5.** Visualization of data points in each stage. The X-axis represents the time cost of the communication process, divided into many small periods. The Y-axis represents the number of data points in a short period.

### 5.2. Experimental Setup

We establish four Gaussian mixture models and train these models with data from different communication stages for estimating their parameters. This way, the distribution of each Gaussian mixture model after data fitting can reflect the normal communication behavior of the corresponding communication stage. For a data point, the well-trained model can predict which Gaussian distribution the data point comes from in a Gaussian mixture model. Then, we can calculate the probability that this data point comes from the predicted Gaussian distribution. We can consider this sample a correct prediction if this probability is larger than the threshold value we set in advance. The ratio of the number of correct predictions to the total number of samples corresponds to the accuracy of this experiment.

The hyperparameters of the Gaussian mixture model are the number of Gaussian distributions owned by the mixture model. In the context of our problem, the model's hyperparameters are also the link number at the corresponding stage. The accuracy of models with different hyperparameters varies. After several rounds of experiments, we found the hyperparameters with the best performance. Table 1 shows our experimental results.

**Table 1.** Highest accuracy and its link number.

|                   | **Model 1** | **Model 2** | **Model 3** | **Model 4** |
|-------------------|-------------|-------------|-------------|-------------|
| Highest accuracy  | 91.56%      | 99.43%      | 97.99%      | 94.07%      |
| Link number       | 13          | 1           | 1           | 38          |

### 5.3. Result Analysis

The results represented in Figure 4 and Table 1 suggest that GMM performs well for some stages since both model 2 and model 3 achieved higher accuracy than with a reasonable link number, while for model 1 and model 4, although they can reach a satisfying

accuracy eventually, the number of links became so large that it became meaningless, especially for the 38 links case. For stages represented by model 2 and model 3, we believe they only use one link to communicate since that is the case the model behaves best. The fact that the number of links for stages represented by model 1 and model 4 is too large to achieve high accuracy suggests that these stages are affected by too many factors that a reasonable Gaussian mixture distribution cannot represent.

As the GMM estimates the parameter values of the model by maximizing the likelihood of the observed data, the resulting model parameters will also be more applicable to the communication scenario that produced the training dataset. However, this does not mean that the model cannot be used in more general anomaly detection scenarios for peer-to-peer communications. When you have a sufficient amount of historical data for a communication scenario, and the communication time of the traffic shows a trend towards more stable data for the most part, it is entirely possible to refit a new GMM model to detect anomalies for that scenario.

## 6. Conclusions

In this work, we proposed a GMM-based traffic analysis method, which assumes that the traffic data comes from Gaussian mixture distribution and uses the learned distribution to predict whether a new data point is behaving normally or under attack. The experimental results suggest that this method works for some but not all communication stages. One possible reason is that these communication stages are affected by too many factors that cannot be represented just in a Gaussian mixture model. Future work may consider organizing these factors in the hierarchy to make the model more reasonable.

Furthermore, the problem that the model proposed in this paper is expected to solve is not limited to the particular dataset used here. The context of the problem only assumes communication between two points via a fixed protocol but does not specify the communication protocol. Future work could therefore validate the validity of the present model using datasets from more realistic scenarios.

**Author Contributions:** Conceptualization, B.Y. and Y.W.; methodology, Y.Z. (Yongzheng Zhang); software, B.Y.; formal analysis, W.X.; investigation, W.X.; data curation, Y.Z. (Yiming Zhao) and W.Z.; writing—original draft preparation, B.Y.; writing—review and editing, Y.W.; supervision, Y.Z. (Yongzheng Zhang); and funding acquisition, Y.W. All authors have read and agreed to the published version of the manuscript.

**Funding:** This research was funded by the National Natural Science Foundation of China (NSFC No. 62272129).

**Data Availability Statement:** Source codes and datasets are available in GitHub repository, and the link is https://github.com/yubinCloud/network-traffic-anomaly-detection-based-on-GMM (accessed on 8 February 2023).

**Conflicts of Interest:** The authors declare no conflict of interest.

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
