# Peer review of "A Network Traffic Anomaly Detection Method Based on Gaussian Mixture Model"

_electronics, doi:10.3390/electronics12061397_

Round 1

Reviewer 1 Report

In general, the topic covered and the methodology discussed is interesting. However, section 3 should be extended by describing the Gaussian mixture model in more detail. Furthermore, the meaning of figure 3 should be better described. Again, the EM algorithm should also be better described, reporting, for example, the mathematical representation of the function Q.

Author Response

Response to Reviewer 1 Comments

Thank you for your guidance and criticism of this thesis. I have revised it to address several of the points you raised and here is my response.

Point 1: Section 3 should be extended by describing the Gaussian mixture model in more detail.

Response 1: I have added two paragraphs in Section 3 to introduce the GMM model, including how the GMM generates data as a generative model and how to use the GMM. The original text has also been revised with care, and additions have been made where the presentation was unclear.

Point 2: The meaning of figure 3 should be better described.

Response 2: I have added more explanation about the figure below the original, clearly explaining each symbol and adding an explanation of the three distributions that appear in the figure.

Point 3: The EM algorithm should also be better described, reporting, for example, the mathematical representation of the function Q.

Response 3: I have added an explanation of the EM algorithm, the key point in the iterative process of the EM algorithm is the Q function, and in this new version I have added a mathematical expression for the Q function and further explained the meaning of the Q function. These changes enrich the interpretation of the EM algorithm and its relevance to the model in this paper.

Reviewer 2 Report

Authors introduce a statistical method for detecting network traffic anomalies using the Gaussian mixture model. They show how to learn the normal behaviour of a communication process under the assumption that its time costs are generated from the Gaussian mixture model. Authors also show that with the learned Gaussian mixture model, they can predict whether a data point is under attack by computing the likelihood that the data point is drawn from the learned Gaussian distribution. The experimental results show that presented method reached a high accuracy in some cases.

The article was written correctly and it raises a very interesting problem. 

I have only a few small remarks that do not detract from the significance of the presented results:

- Please add some spaces before references in text. For example: In terms of statistical methods, Callegari[7] et al proposed … It should be: In terms of statistical methods, Callegari [7] et al proposed.

- When you add reference to the figure you use Fig1. If should be Fig. 1.

In the article, I missed the section on the results of considering other connection scenarios.

I missed the question of how the accuracy of the proposed methods changes depending on the type of connection and the structure of the transmitted traffic in the network.

Are there any general conclusions regarding accuracy based on the scenario proposed in the paper?

I suggest accepting the article for publication after addressing my comments and supplementing minor deficiencies.

Author Response

Response to Reviewer 2 Comments

Thank you for your guidance and criticism of this thesis. I apologize for any omissions in my previous work. I have revised it to address several of the points you raised and here is my response.

Point 1: Please add some spaces before references in text. When you add reference to the figure you use Fig1. If should be Fig. 1.

Response 1: I am very sorry for this small oversight in the formatting. We have fixed these issues in the new version.

Point 2: Are there any general conclusions regarding accuracy based on the scenario proposed in the paper?

Response 2: We have added to the conclusions based on the context of the problem and the data set used for the experiment.

In fact, the model proposed in this paper does not impose restrictions on the connection protocol for the desired solution to the problem of anomaly detection of network connections between two points through traffic communication times. The dataset used for our experiments contains six communication stages containing four protocols, and we build four Gaussian mixture models to validate our ideas separately. The experimental results show that the GMM models all achieve a high level of accuracy.

In addition to this, we also explain in the future work section that more connection protocols need to be verified.

Round 2

Reviewer 1 Report

I thank the authors for the work done. The paper is clear enough now.